# High-Performance and Flexible Design Scheme with ECC Protection in the Cache

**DOI:** 10.3390/mi13111931

**Published:** 2022-11-09

**Authors:** Yulun Zhou, Hongxia Liu, Qi Xiang, Chenyu Yin

**Affiliations:** Key Laboratory for Wide Band Gap Semiconductor Materials and Devices of Education, School of Microelectronics, Xidian University, Xi’an 710071, China

**Keywords:** ECC, cache, reliability, error protection

## Abstract

To improve the reliability of static random access memory (SRAM), error-correcting codes (ECC) are typically used to protect SRAM in the cache. While improving the reliability, we also need additional circuits to support ECC, including encoding and decoding logic. In a high-speed circuit such as a CPU, the L1 cache maintains the same frequency as the CPU, and the decoding of the ECC codes in the cache consumes considerable combinational logic, resulting in limited frequency and performance. This study proposes a high-performance and flexible design scheme with ECC protection in the cache, in which the cache has two working modes: a high-performance mode and a high-reliability mode. The high-performance mode uses simple ECC codes, which can maintain high frequency with low access latency. The high-reliability mode uses more complex ECC codes, which improves the error correction capability and enhances the reliability of the SRAM. To meet the application requirements of different scenarios, the proposed scheme supports the software in switching between the above two modes by configuring the register, which improves the flexibility of the system. The results of synthesis show that the theoretical maximum frequency of proposed ECC design scheme increased from approximately 1.4 GHz in the conventional ECC design scheme to approximately 2.2 GHz. Some of the error correction capability of the high-performance mode is traded off against a 57% increase in frequency. In the high-reliability mode, the error correction capability of the SRAM is enhanced; however, the latency of accessing the cache increases by one cycle.

## 1. Introduction

Reliability is one of the most important considerations in microprocessor design. In today’s microprocessors, approximately 60% of the on-chip area is occupied by the cache [1]. SRAM is an important part of the cache; however, because of the active feedback in the cross-coupled inverter pair, it can easily appear as a single-event upset (SEU) [2]. In particular, data errors are more likely to occur in the case of low voltages [3]. However, reducing the operating voltage is a common method used by current processors to reduce power consumption; therefore, it is necessary to consider the reliability of SRAM in the cache.

ECC is a commonly used protection method for SEU in SRAM, and usually includes parity and Single-Error Correction–Double-Error Detection (SEC–DED) code schemes [4,5,6]. SEC–DED is more widely used in cache because of its superior error correction and detection capabilities; however, SEC–DED also requires more complex encoding and decoding logic than parity, which increases circuit delay and easily limits the frequency of the cache. The cache needs tags and data to achieve its basic functions, and both tags and data are stored in SRAM. In the conventional ECC design scheme, after reading the tags and data from the SRAM, it directly passes through the ECC decoder, compares the corrected tag, and selects the corresponding data. The delay in this procedure is relatively large, which limits the frequency of the circuit. Compared with the conventional scheme using SEC–DED protection for both tags and data, this study proposes a high-performance pipeline design scheme with ECC in the cache, which uses SEC–DED to protect data but only detects errors of tags without correcting them. We use the TSMC 12 nm library for logic synthesis, and the results show that, after sacrificing a small part of the error correction ability, the theoretical maximum frequency of this data path increased from approximately 1.4 GHz in the conventional scheme to approximately 2.2 GHz, resulting in an approximate frequency increase of 57%. In addition, when the chip faces high-energy particles in the aerospace environment or the circuit enters low-voltage mode, it increases the possibility of SEU in the SRAM. Therefore, we also have higher reliability requirements in certain scenarios. With the scaling of process technology, the probability of multi-bit upsets (MBUs) is also increasing, and most MBUs occur in adjacent bits [7]. Therefore, in addition to the above-mentioned high-performance circuit design scheme, this study also proposes a high-reliability circuit design scheme. In this scheme, SEC–DED is used to protect the tag and Single Error Correction–Double Error Detection–Double Adjacent Error Correction (SEC–DED–DAEC) proposed in [8] is used to protect the data. The high-reliability mode improves error correction and can maintain the same frequency as the high-performance mode; however, the latency of accessing the cache increases by one cycle. There are different requirements for performance and reliability in different application scenarios. This solution supports switching between the above two modes by configuring an extended custom field in the control and status register (CSR), which improves the flexibility of the system.

## 2. Related Work

To enhance the reliability of the microarchitecture, Zhang proposed a replication cache scheme that adds a small full associative cache to store a replica of every write to the cache [9]. Paul et al. proposed a reliability-driven ECC allocation scheme that matches the relative vulnerability of a memory block with appropriate ECC protection [10]. Manoochehri et al. proposed a new reliable write-back cache that adds correction capability to parity protection [1]. Farbeh et al. proposed an architecture that extends the data-protection granularity from a single word to multiple words by exploiting the parallel cache lines access [11].

To improve the error detection and correction capability of ECC codes, Sánchez-Macián et al. proposed SEC–DAEC and SEC–DED–Triple Adjacent Error Detection (SEC–DED-TAED) coding schemes [12]. Azad et al. proposed a SEC–DED–DAEC coding scheme [8], and this study uses that scheme.

## 3. Design Schemes

### 3.1. Parameter and Configuration

We assume that this module is a L1 data cache. The main parameters and configuration are shown in Table 1, where the cache size is 64 KB, the cache line is 64 bytes, 4-way set-associative, and the address width of the CPU is 40-bit. After calculation, it can be concluded that there are 256 sets, that the index width required by the index set is 8-bit, and that the offset of the cache line is 6-bit. Therefore, the width required for the tag is 26-bit, and the corresponding SEC–DED parity bit width is 6-bit. Because the cache supports multi-port access, a multi-bank structure is used, and the data array of each way is divided into eight banks; the data of each bank is 64-bit, and the corresponding ECC check bit width is 8-bit. Therefore, the Tag SRAM in each way is 832 B (256 × 26 bits), the Tag ECC register file is 192 B (256 × 6 bits), the Data Bank SRAM is 2 KB (256 × 64 bits), and the Data ECC register file is 256 B (256 × 8 bits). We temporarily ignore the storage of the cache coherence state and other features.

### 3.2. Conventional Design Scheme

A conventional pipeline scheme with ECC is shown in Figure 1. After the tags and corresponding ECC check bits are read from the tag array as the input of the ECC decoder, the single-bit error in the tags can be repaired and the double-bit error can be detected. After processing by the ECC decoder, the corrected tag is compared with the request address, and the cache hit or miss is judged and passed to the next stage through D flip-flop. The data and corresponding ECC check bits are read from the data array. Because there are multiple banks in each way, it is necessary to select the read data to obtain the required data and ECC check bits and input them into the ECC decoder. Similar to the processing of the tag, error correction and error detection will also be performed on the data and the error-corrected data will be transmitted to the next stage through D flip-flop. In the next cycle, the data are selected based on the results of the tag comparison, and if it is a cache hit, the data of the corresponding way is returned. If the ECC decoder finds a double-bit error, it reports an exception.

The combinational logic delay in the conventional design scheme is too large, which limits the frequency and performance of the cache. The specific results will be presented in Section 4.

### 3.3. Proposed High-Performance Design Scheme

In the conventional design scheme, reading the tag or data from the SRAM requires a large delay in logic. If the ECC decoder is placed at this stage, the frequency of the cache will be limited. For a high-performance CPU, frequency is crucial to performance; therefore, certain improvements can be made to pipeline schemes.

Figure 2 shows the high-performance design scheme proposed in this study. To optimize the timing, we move the ECC decoder to the next stage, still use the uncorrected tag to compare with the request address when judging cache hits or misses, and pass the read tag and the corresponding ECC check bits to the next stage. In the next stage, the tag and ECC check bits of the hit way are selected to enter the ECC decoder for error detection, which also saves the area of another three ECC decoders. The tag comparison was completed at the previous stage; therefore, the ECC decoder only detects errors without correcting them, which is equivalent to Single Error Detection–Double Error Detection (SED–DED). 

The tag ECC decoder is specifically optimized in this solution, and a signal tag_error is added to indicate whether the tag has errors. The signal is asserted only when the stored tag contains an error. If the ECC check bits are flipped, tag_error will not be asserted, which improves the efficiency of the system. We consider 4-bit data and 4-bit corresponding ECC check bits as examples. After the data and ECC check bits were used as the input of the ECC decoder, as shown in Table 2, a 4-bit syndrome signal was calculated by the XOR logic of the corresponding bits inside the decoder. As shown in Table 3, the tag_error signal is asserted only when a double-bit error or a single-bit error occurs in the position of the data. The tag in the cache is 26-bit, and the corresponding ECC check bits are 6-bit. The tag_error signal is asserted only when the bit in the tag is flipped. In this case, if it is originally judged to be a cache hit, it is actually a miss, and an external request must be initiated through the bus to obtain the cache line corresponding to the requested address.

Reading data from the data array consumes a lot of time; therefore, to optimize the timing, the data and ECC check bits of the corresponding bank is directly sent to the next stage after the bank selecting. In the next stage, the data and the corresponding ECC check bits read from the register are checked using the SEC–DED decoder. Herein, the ECC decoder is different from that for tag, which can correct the data with single-bit error. If a double-bit error occurs in the data, it is handled by reporting the corresponding exception. The corrected data are selected using the result of the tag comparison, and the data of the hit way are returned. The final returned data are determined by the current ECC mode, which is described in detail in Section 3.5.

In addition to requests from instructions like load and store, writeback and snoop also need to access the data array in order to read the cache line. The combinational logic delay will be too long if the read data is returned directly after error correction, resulting in a limited frequency. In high-performance mode, for better timing considerations, the data is read from the data array and returned directly without error correction, and the correctness of the data is checked in the next cycle. If the data is incorrect, an exception will be reported.

The key point of the high-performance design scheme is to sacrifice a small amount of tag reliability in exchange for a larger frequency.

### 3.4. Proposed High-Reliability Design Scheme

In some special application scenarios, such as aerospace applications or the low-voltage mode of circuits, the probability of SEU occurring in SRAM is significantly improved. To ensure the correctness of the data and the stability of the system, ECC coding with a stronger error correction capability can be used.

As shown in Figure 3, a high-reliability circuit design scheme proposed in this study. Compared with the convention design scheme, the error-correction capability of this scheme is improved, and adjacent 2-bit errors of data can be corrected.

In the case of the tag, the protection method of SEC–DED is used, and the tag and corresponding ECC check bits are read out in stage 0. In stage 1, they are checked, and the errors are corrected. The error-corrected tags are then compared with a request address to determine cache hits ot misses. In stage 2, based on the obtained results from stage 1, the corresponding way in the data pipeline is selected or other modules like performing store operation or handling cache misses are requested. Single-bit errors in the tag can be corrected, and an exception will be reported if a double-bit error is detected.

As mentioned in Section 1, MBUs mostly occur on adjacent bits [7], so choosing an ECC code that can correct adjacent 2-bit errors with less overhead is more cost effective than choosing an ECC code that can correct any 2-bit errors. This study uses the SEC–DED–DAEC scheme proposed in [8] for the ECC protection of data, which can correct 1-bit and adjacent 2-bit errors and detect other uncorrectable errors.

The H-matrix of the SEC–DED–DAEC is shown in Figure 4. The decoding logic of this scheme is more complex than that of SEC–DED. Therefore, to keep the high frequency of the circuit unaffected, the decoding in the design scheme is changed to two cycles. In the first cycle, the syndrome is calculated by the XOR operation of the bits that are one in each row of the H-matrix. In the second cycle, the syndrome is compared with the value of each column in the H-matrix to detect and correct errors. The SEC–DED–DAEC decoder will output corrected data or report uncorrectable errors eventually. An exception is reported if an uncorrectable error occurs, otherwise the corrected data are selected according to the result of the tag comparison and then returned, and the final returned data are also determined by the current ECC mode. For writeback and snoop requests, the data read from the data array also needs two cycles to return the error-corrected data. If an uncorrectable error occurred in the data, it was also handled through the corresponding exception.

Compared with the high-performance design scheme, the latency of accessing the cache in this scheme increases by one cycle; however, the error-correction ability is improved.

### 3.5. Flexible Design Scheme

The above two design schemes are two working modes of the cache actually, and the corresponding modules in the two schemes are integrated into the final cache as two paths. Therefore, our flexible design scheme already includes the high-performance and high-reliability schemes. As shown in Figure 5, only one SEC–DED check code needs to be stored for the tag; however, the tag will not be corrected in the high-performance mode. For the data, there are two ECC codes that must be stored: SEC–DED and SEC–DED–DAEC. The tags and data both have two paths, one for each mode; the upper one is the high-performance mode and the lower one is the high-reliability mode. Finally, the MUX will select the control information and data of the corresponding path according to the current ECC mode. The high-reliability mode has superior error correction ability than that of the high-performance mode; however, the latency of accessing the cache will increase by one cycle.

For low power consumption, only the code array corresponding to the current ECC mode is read and written, and all cache lines need to be invalidated when the mode is switched. If there is a dirty cache line, it needs to be written back to the lower-level cache. Because the code array corresponding to the new ECC mode does not store the latest data encoding, an unexpected ECC error will occur when the data is read again at this time.

Working-mode switching was achieved by controlling the selection of the MUX using the custom ecc_mode field in the CSR register. Different values can be configured for this field of the CSR register according to the actual scenarios and application requirements, which realizes the high flexibility of the system.

## 4. Results of Synthesis

We used the Synopsys Design Compiler to synthesize the above circuit under the TSMC 12 nm library (slow process corner), where the SRAM uses low V threshold (lvt) cells. The following tables present the results of the synthesis, which are the tag pipeline and data pipeline of the conventional design scheme and proposed design scheme, respectively.

The delays of the tag read path and data read path (stage 0) of the proposed design scheme are equivalent to subtracting the delay of the ECC decoder in the conventional design scheme from the total delay, which are 0.34 ns and 0.32 ns, respectively. The following tables list the timing results of the proposed design scheme at the next stage (stage 1) of the tags and data read.

We can see that, in the high-performance path, the delay of the tag read path and data read path is close to that of the next stage pipeline, which is ideal for pipeline design. In the conventional design scheme, as can be seen from Table 4 and Table 5, the timing of the critical path is 0.60 ns of the data read path; we assume that 15% of the clock uncertainty and the setup time of approximately 0.02 ns are reserved, and the maximum frequency is approximately 1.4 GHz. As it can be seen from Table 6 and Table 7, in the high-performance path, the timing of the critical path is 0.38 ns of the next stage of the data read, and with the same constraints, the highest frequency is approximately 2.2 GHz. At the same time, the area of the three ECC decoders was also saved.

We examine the timing of the high-reliability path. In stage 2 of the tag pipeline, the store operation needs to use the original stored data, so the corrected data needs to be used when requesting other modules. The delay of this timing path is 0.32 ns. According to the timing reports of the previous two schemes, because there are many identical modules, it can be inferred that the critical path of the tag pipeline is stage 1, which is approximately 0.36 ns, and will not become the critical path of the cache. For the data pipeline, owing to the more complicated combinational logic of the SEC–DED–DAEC decoder, a pipeline method was used in the design to increase the frequency. As shown in Table 8 and Table 9, the first stage in the SEC–DED–DAEC decoder completes the calculation of the syndrome, and the total delay of this path is 0.33 ns. The second stage completes error detection and correction according to syndrome and outputs corrected data or reports the error. The total delay of this path was 0.26 ns, which was smaller than the delay of the critical path in the high-performance path. Thus, it can operate at the same frequency as the high-performance mode.

Figure 6 shows the delay at each stage of the high-performance and high-reliability paths. Because the access latency of the high-performance path is one cycle less than that of the high-reliability path, the high-performance path does not have the delay of stage 2. The critical path of the flexible design scheme is data pipeline stage 1 of the high-performance path, with a delay of 0.38 ns.

We have synthesized the design under different configurations, and the area results obtained from the synthesis are shown in Table 10. Additional SRAM is required to store the corresponding ECC check bits when supporting ECC, so a larger area is also required. Compared with the design that only supports high-performance mode, a flexible design scheme support two working modes and has higher reliability to choose from. An area overhead of approximately 22.5% is acceptable.

## 5. Conclusions

Compared with the conventional scheme that uses SEC–DED protection for both tags and data in the cache, the proposed high-performance design scheme in this study has the ability to detect errors for tags and can protect data with SEC–DED at the same time. However, the probability of SEU occurrence under normal scenarios is relatively low because the width of the tag is not large. Sacrificing reliability to a limited extent results in a large frequency increase, which is acceptable for application scenarios that pursue performance. At the same time, the proposed high-reliability design scheme can more effectively correct 1-bit and 2-bit errors compared to the traditional scheme for correcting 1-bit error in the data while meeting the needs of some special high-reliability application scenarios. Finally, the cache proposed in this study can meet the needs of different application scenarios because switching between different operating modes is supported and therefore can improve the flexibility of the system.

## Figures and Tables

**Figure 1 micromachines-13-01931-f001:**
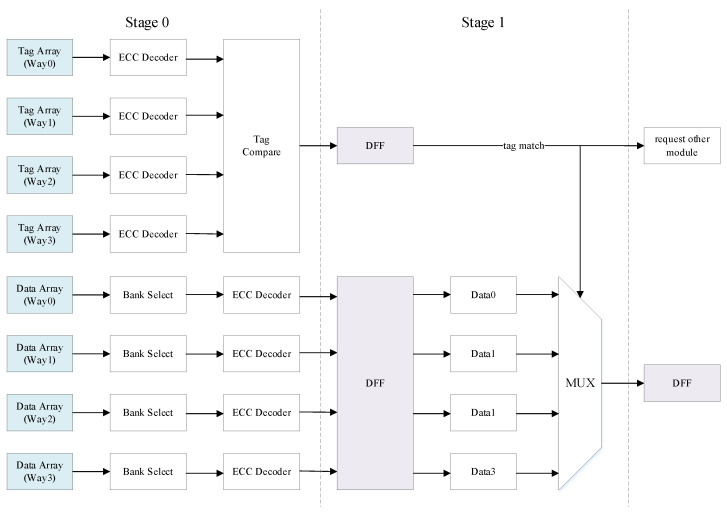
Conventional pipeline design scheme with ECC protection in cache.

**Figure 2 micromachines-13-01931-f002:**
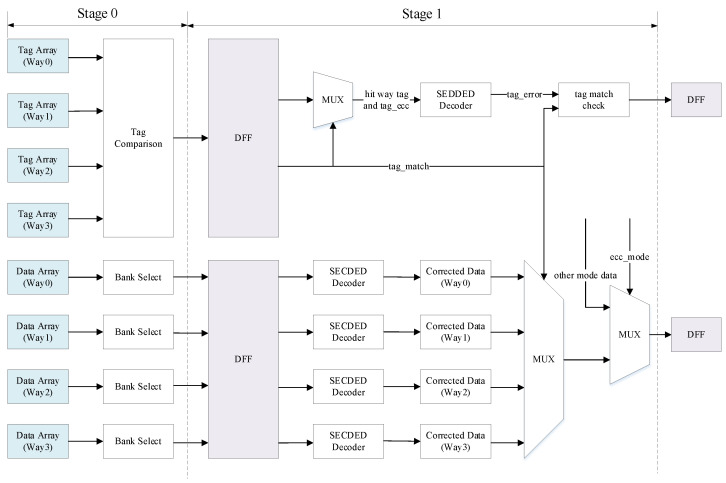
Proposed high-performance design scheme with ECC protection in the cache.

**Figure 3 micromachines-13-01931-f003:**
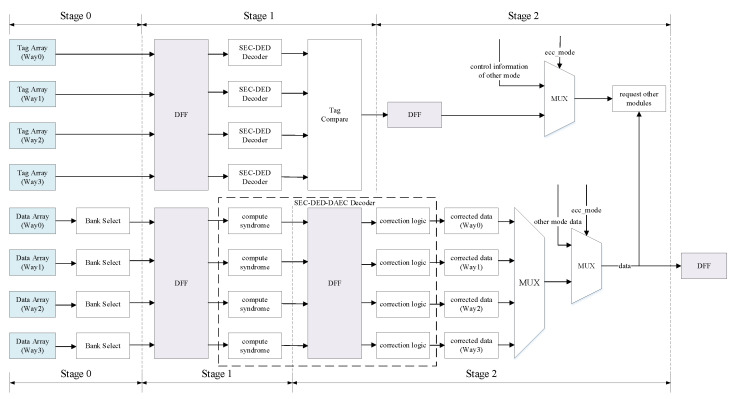
Proposed high-reliability design scheme with ECC protection in cache.

**Figure 4 micromachines-13-01931-f004:**
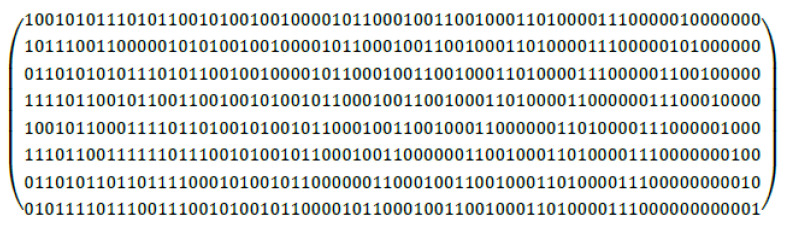
H-matrix for SEC–DED–DAEC (72, 64) code [8].

**Figure 5 micromachines-13-01931-f005:**
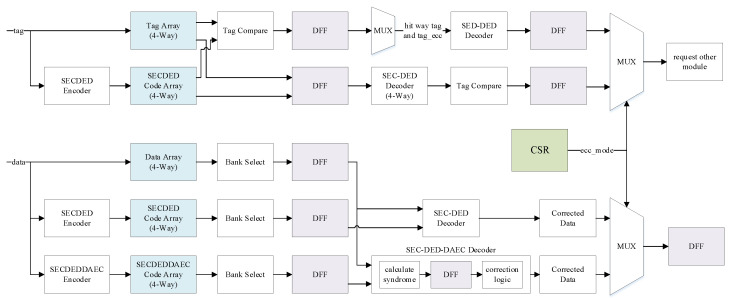
Flexible design scheme with ECC protection in cache.

**Figure 6 micromachines-13-01931-f006:**
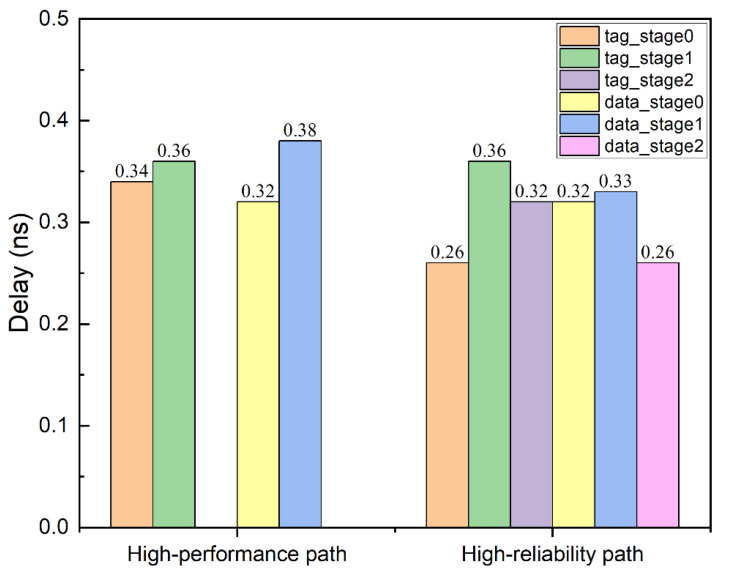
Delay at each stage of the two paths.

**Table 1 micromachines-13-01931-t001:** Cache parameter and configuration.

Parameter	Value
CPU address width	40 bits
Cache size	64 KB
Cache line size	64 Bytes
Way number	4
Set number	256
Tag width	26 bits
Bank number	8
Bank size	64 bits

**Table 2 micromachines-13-01931-t002:** Syndrome calculation logic.

No.	1	2	3	4	5	6	7	8
bit	ecc[0]	ecc[1]	data[0]	ecc[2]	data[1]	data[2]	data[3]	ecc[3]
syndrome[0]	x		x		x		x	
syndrome[1]		x	x			x	x	
syndrome[2]				x	x	x	x	
syndrome[3]	x	x	x	x	x	x	x	x

**Table 3 micromachines-13-01931-t003:** Error detection logic.

Syndrome	Error Bit	Single Error	Double Error	Tag Error
0000	No error	0	0	0
1000	ecc[4]	1	0	0
1001	ecc[0]	1	0	0
1010	ecc[1]	1	0	0
1011	data[0]	1	0	1
1100	ecc[2]	1	0	0
1101	data[1]	1	0	1
1110	data[2]	1	0	1
1111	data[3]	1	0	1
other	double error	0	1	1

**Table 4 micromachines-13-01931-t004:** Tag read path timing of the conventional design scheme.

Logic	Delay (ns)
Read from tag array	0.26
SEC–DED decoder (26-bit)	0.22
Tag compare	0.08
Total	0.56

**Table 5 micromachines-13-01931-t005:** Data read path timing of the conventional design scheme.

Logic	Delay (ns)
Read from data array	0.26
Data select	0.06
SEC–DED decoder (64-bit)	0.28
Total	0.60

**Table 6 micromachines-13-01931-t006:** Tag read path stage 1 timing of the high-performance path.

Logic	Delay (ns)
Read from register	0.06
MUX (select tag match way)	0.05
Tag ECC decoder (26-bit, error detection only)	0.18
Tag match check	0.07
Total	0.36

**Table 7 micromachines-13-01931-t007:** Data read path stage 1 timing of the high-performance path.

Logic	Delay (ns)
Read from register	0.06
SEC–DED decoder (64-bit)	0.28
MUX (select data)	0.04
Total	0.38

**Table 8 micromachines-13-01931-t008:** First-stage timing of the SEC–DED–DAEC decoder.

Logic	Delay (ns)
Read from register	0.06
Compute syndrome	0.27
Total	0.33

**Table 9 micromachines-13-01931-t009:** Second-stage timing of the SEC–DED–DAEC decoder.

Logic	Delay (ns)
Read from register	0.06
Correction logic	0.16
MUX (select data)	0.04
Total	0.26

**Table 10 micromachines-13-01931-t010:** Area results for a different design scheme.

Scheme	No ECC Protection	High-Performance Path Only	Flexible Design Scheme
Area (μm^2^)	130,540	173,664	212,698

## Data Availability

Not applicable.

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
