# Peer review of "High-Performance and Flexible Design Scheme with ECC Protection in the Cache"

_micromachines, 2022, doi:10.3390/mi13111931_

Round 1

Reviewer 1 Report

1.       The abstract is clear and written in simple English.

2.       Page no.2, lines 48 to 52. Hard to read the sentence.

This study proposes a high- 46 performance pipeline design scheme with ECC in the cache, using the TSMC 12 nm library 47 for logic synthesis; the results show that after sacrificing a small part of the error correction 48 ability (for tag, errors can only be detected but not corrected; for data, normal SEC-DED 49 is used), the theoretical maximum frequency of this data path increased from approxi- 50 mately 1.4 GHz in the traditional scheme to approximately 2.2 GHz, resulting in an ap- 51 proximate frequency increase of 57% and saving of some area.

I recommend elaborating on how much area saving has been done, and the obtained result must be compared with any existing method.

3.       The author mentioned the conventional pipeline design scheme in Figure1. However, there is no comparison with the conventional method. Add any existing work using this scheme, and the results must be compared with the proposed one.

4.       In line no. 231 author claims, "The second stage completes the work on error correction, error detection, and output data." However, no scheme or method is proposed and not evaluated for error detection and correction.

5.       Lines 257 to 266 are instructions for the references, which may be removed.

6.       The reference list is not recent and only the conference paper. Need to modify it with the latest journal article.

7.       Add some of the related and recent references to compare your study. 

Author Response

Dear reviewer,

We would like to thank you for your efforts in reviewing our manuscript titled "High-performance and Flexible Design Scheme with ECC Protection in Cache", and providing many helpful comments and suggestions, which are valuable for the revision and improvement of our study.

We revised the abstract to make it simple to understand, added Related Work in Chapter 2, wrote more detailed description of design scheme, added figure to compare timing results, added area results of synthesis, etc.

Thank you again for your comments and suggestions.

Response to Reviewer Comments:

Point 1: The abstract is clear and written in simple English.

Response 1: Thank you for your constructive comments. We revised the abstract to make it simple to understand.

Point 2: Page no.2, lines 48 to 52. Hard to read the sentence.

This study proposes a high- 46 performance pipeline design scheme with ECC in the cache, using the TSMC 12 nm library 47 for logic synthesis; the results show that after sacrificing a small part of the error correction 48 ability (for tag, errors can only be detected but not corrected; for data, normal SEC-DED 49 is used), the theoretical maximum frequency of this data path increased from approxi- 50 mately 1.4 GHz in the traditional scheme to approximately 2.2 GHz, resulting in an ap- 51 proximate frequency increase of 57% and saving of some area.

I recommend elaborating on how much area saving has been done, and the obtained result must be compared with any existing method.

Response 2: Thank you for your advice. We have rewritten this part for easier to read, and delete area related words, because it will be elaborated on later chapter.

Point 3: The author mentioned the conventional pipeline design scheme in Figure1. However, there is no comparison with the conventional method. Add any existing work using this scheme, and the results must be compared with the proposed one.

Response 3: Thank you for your constructive comments. The work of comparing conventional scheme with proposed one is in Chapter 4. It’s inappropriate to ingore it indeed, so we added some sentences here to indicate that comparison will be presented in Chapter 4.

Point 4: In line no. 231 author claims, "The second stage completes the work on error correction, error detection, and output data." However, no scheme or method is proposed and not evaluated for error detection and correction.

Response 4: Thank you for your constructive comments. This is SEC-DED-DAEC decoder, we added some details of scheme here, and it is described detailedly in Section 3.4.

Point 5: Lines 257 to 266 are instructions for the references, which may be removed.

Response 5: Thank you for your constructive comments. The instrucitons have been removed.

Point 6: The reference list is not recent and only the conference paper. Need to modify it with the latest journal article.

Response 6: Thank you for your constructive comments. We added related work in Chapter 2, and there are more references of recent journal article.

Point 7: Add some of the related and recent references to compare your study. 

Response 7: Thank you for your constructive comments. We added more related and recent references.

Reviewer 2 Report

1. Please add comparison table with other work. It is difficult to figure out that the improvement of this work is superior or not compared to other schemes.

2. Please re-organize synthesis results. It will be better to include comparison chart rather than table.

3. If possible, please include place and route results. This will be helpful to understand the area advantage (or overhead) of the proposed scheme. 

4. For the high-performance and high-reliability design schemes, more detailed description is needed. Many readers will feel difficult to understand the key points of the two proposed structures.

5. Please add the synthesis results of the flexible design scheme.  Although the core scheme is the same with high-performance and high-reliability schemes, the added block may affect the total delay. So, it will be good to show the synthesis results of the flexible design.

Author Response

Dear reviewer,

We would like to thank you for your efforts in reviewing our manuscript titled "High-performance and Flexible Design Scheme with ECC Protection in Cache", and providing many helpful comments and suggestions, which are valuable for the revision and improvement of our study.

We revised the abstract to make it simple to understand, added Related Work in Chapter 2, wrote more detailed description of design scheme, added figure to compare timing results, added area results of synthesis, etc.

Thank you again for your comments and suggestions.

Response to Reviewer Comments:

Point 1: Please add comparison table with other work. It is difficult to figure out that the improvement of this work is superior or not compared to other schemes.

Response 1: Thank you for your constructive comments. Because commercial CPU companies rarely publish specific implementation details, we are difficult to find the data to compare. However, the CPU that can protect the tag in the cache by SEC-DED must be similar to the conventional design scheme in this article. We compared the frequency of the proposed design scheme with that of conventional design scheme, and added Table 10 to compare the area between different schemes.

Point 2: Please re-organize synthesis results. It will be better to include comparison chart rather than table.

Response 2: Thank you for your advice. We added Figure 6 to describe the delay at each stage of the high-performance and high-reliability paths.

Point 3: If possible, please include place and route results. This will be helpful to understand the area advantage (or overhead) of the proposed scheme. 

Response 3: Thank you for your advice. We are sorry that we focus on digital IC front-end design and are not familiar with bank-end flow. However, there are also area report from the synthesis and we added Table 10 to show the area between different schemes.

Point 4: For the high-performance and high-reliability design schemes, more detailed description is needed. Many readers will feel difficult to understand the key points of the two proposed structures.

Response 4: Thank you for your advice. We added more detailed description in the high-performance and high-reliability design schemes.

Point 5: Please add the synthesis results of the flexible design scheme.  Although the core scheme is the same with high-performance and high-reliability schemes, the added block may affect the total delay. So, it will be good to show the synthesis results of the flexible design.

Response 5: Thank you for your constructive comments. Flexible design scheme consists of high-performance and high-reliability schemes actually. We may not describe this clearly in the previous version, so we added some sentences to highlight this in Section 3.5. The synthesis results of the two schems are the corresponding paths in the flexible design scheme actually, so the total delay already considers the added blocks. We added Figure 6 to describe the delay at each stage of the two paths.

Round 2

Reviewer 2 Report

The manuscript has been revised properly according to the comments.